# Novel Benzothiazole-Based Ureas as 17β-HSD10 Inhibitors, A Potential Alzheimer’s Disease Treatment

**DOI:** 10.3390/molecules24152757

**Published:** 2019-07-29

**Authors:** Laura Aitken, Ondrej Benek, Brogan E. McKelvie, Rebecca E. Hughes, Lukas Hroch, Monika Schmidt, Louise L. Major, Lucie Vinklarova, Kamil Kuca, Terry K. Smith, Kamil Musilek, Frank J. Gunn-Moore

**Affiliations:** 1University of St. Andrews, School of Biology, Medical and Biological Sciences Building, North Haugh, St. Andrews KY16 9TF, UK; 2University Hospital, Biomedical Research Center, Sokolska 581, 500 05 Hradec Kralove, Czech Republic; 3University of Hradec Kralove, Faculty of Science, Department of Chemistry, Rokitanskeho 62, 500 03 Hradec Kralove, Czech Republic; 4Cancer Research UK Edinburgh Centre, MRC Institute of Genetics and Molecular Medicine, Western General Hospital, University of Edinburgh, Edinburgh EH4 2XU, UK; 5Biomedical Science Research Complex, University of St. Andrews, North Haugh, St. Andrews KY16 9ST, UK

**Keywords:** Alzheimer’s disease (AD), amyloid-beta peptide (Aβ), mitochondria, 17β-hydroxysteroid dehydrogenase type 10 (17β-HSD10), amyloid binding alcohol dehydrogenase (ABAD), benzothiazole

## Abstract

It has long been established that mitochondrial dysfunction in Alzheimer’s disease (AD) patients can trigger pathological changes in cell metabolism by altering metabolic enzymes such as the mitochondrial 17β-hydroxysteroid dehydrogenase type 10 (17β-HSD10), also known as amyloid-binding alcohol dehydrogenase (ABAD). We and others have shown that frentizole and riluzole derivatives can inhibit 17β-HSD10 and that this inhibition is beneficial and holds therapeutic merit for the treatment of AD. Here we evaluate several novel series based on benzothiazolylurea scaffold evaluating key structural and activity relationships required for the inhibition of 17β-HSD10. Results show that the most promising of these compounds have markedly increased potency on our previously published inhibitors, with the most promising exhibiting advantageous features like low cytotoxicity and target engagement in living cells.

## 1. Introduction

There is a strong, well-documented connection between Alzheimer’s disease (AD) and mitochondrial dysfunction [1,2,3]. Mitochondrial changes in AD patients are an early event, preceding the onset of amyloid plaque formation, and include morphology abnormalities and changes in metabolism stemming from alterations in the complexes of the electron transport chain, the enzymes in the tricarboxylic acid cycle (TCA), and changes in components of the mitochondrial membrane involved in import/export flux. Mitochondria are key to the production of adenosine triphosphate (ATP) via the metabolism of glucose and fatty acids. Mitochondrial dysfunction in AD includes activity changes in many enzymes involved in these processes and contributes to the reduction in energy metabolism in AD [4]. Mitochondrial dysfunction is also exacerbated by the presence of amyloid beta peptide (Aβ) within mitochondria [5]. One mitochondrial enzyme affected in AD is 17β-HSD10 (17β-hydroxysteroid dehydrogenase type 10, also known as amyloid-β binding alcohol dehydrogenase (ABAD) or 3-Hydroxyacyl-CoA dehydrogenase). 17β-HSD10’s primary role is to utilise several substrates to produce energy in the β-fatty acid oxidation pathway, the energy source when glucose levels are low, playing a prominent role in AD, where glucose metabolism is significantly decreased [6]. Importantly, we and others have shown that inhibition of this enzyme is beneficial in both in vitro and in vivo AD models in its own right and also protects against Aβ toxicity in both cellular and transgenic mouse models of AD [7,8,9,10,11,12]. A current working hypothesis is that by inhibiting the enzyme activity of 17β-HSD10 (a contributor to the β-fatty acid oxidation pathway) this can help re-balance alterations in glucose metabolism observed in AD (Aitken unpublished data). 

17β-HSD10 was first identified as an Aβ binding protein in 1997 [13], a finding which has subsequently been confirmed using a number of techniques [5,13,14]. 17β-HSD10 is known to interact with the two major plaque forming isoforms of Aβ, namely Aβ(1–40) and Aβ(1–42), leading to distortion of the enzyme structure and inhibition of its normal function as an energy provider for cells [15,16]. In vitro experiments have shown that the interaction between 17β-HSD10 and Aβ is cytotoxic and 17β-HSD10’s function is altered with a build-up of reactive oxygen species (ROS) and toxins leading to mitochondrial dysfunction. Using site-directed mutagenesis and surface plasmon resonance protein interaction assays (SPR), Lustbader et al. identified the L_D_ loop of the 17β-HSD10 protein as the binding site for Aβ and subsequently synthesised a 28-amino acid peptide encompassing this region, which was termed the 17β-HSD10 decoy peptide [5]. Using SPR assays it has been shown that this 17β-HSD10 decoy peptide can prevent the binding of 17β-HSD10 to Aβ(1–40) and Aβ(1–42). Significantly, inhibition of the interaction between 17β-HSD10 and Aβ by the 17β-HSD10-decoy peptide was shown to translate into a cytoprotective effect in cell culture experiments. Cortical neurons exposed to Aβ(1–42) showed a significant increase in cell death, as measured by cytochrome-c release, whilst those pre-incubated with the 17β-HSD10 decoy peptide did not. Critically, for the first time, this work demonstrated that inhibition of the 17β-HSD10-Aβ interaction may target potential disease- relevant mechanisms. 

Other than the disruption of the 17β-HSD10/Aβ interaction, there is a second approach which may hold merit in treating AD: the direct modulation of 17β-HSD10 enzyme activity. In vitro experiments with neuronal-like SHSY-5Y cells exposed to the 17β-HSD10 inhibitor AG18051, showed a reduction in mitochondrial dysfunction and oxidative stress associated with the interaction between 17β-HSD10 and Aβ and protected the cells from Aβ-mediated cytotoxicity [7,8]. This proved that inhibiting 17β-HSD10 activity may also be a viable therapeutic approach for the treatment of AD. 

In our previously published work [9] we discuss the rationale behind utilising analogues of the FDA-approved drugs frentizole and riluzole as inhibitors of 17β-HSD10 for potential therapeutics in AD. Briefly, many benzothiazole analogues have been shown to possess various biological activities in the central nervous system, with riluzole itself highlighted as neuroprotective. Thus, we focused on generating potent 17β-HSD10 inhibitors based on the benzothiazole scaffold, identifying several potent inhibitors (Figure 1) [9].

These compounds highlighted key structural features required for 17β-HSD10 inhibition with the 6-trifluromethoxy and 6-halogen substitution of the benzothiazole moiety and 3-chloro, 4-hydroxy substitution of the phenyl moiety proving the most favourable, however, with limited solubility the compounds were not optimal for cellular evaluation. The aim of this study was not only to generate benzothiazole urea scaffolds which would show improved potency, but to also generate compounds with improved tolerance and less cytotoxicity within our cellular assays, i.e., better pharmacokinetic parameters. To that end, four series of compounds have been synthesised targeting the key areas of benzothiazole moiety, phenyl ring and urea linker (Figure 2). 

## 2. Results and Discussion

### 2.1. Structural Design and Chemical Synthesis

The first series of compounds is a continuation of our previously reported work [9]. While the benzothiazole scaffold and urea linker were kept intact, further substitution changes into the distal phenyl ring were introduced, mainly at position 3 (Table 1). Methoxy substitutions at position 6 of benzothiazole were selected and based on the comparable inhibitory activity with halogenated analogues and due to the availability of starting material and improved physical chemical properties.

Benzothiazolylureas were formed using the two-step reaction process subsequently described. Initially, 6-methoxybenzo[*d*]thiazol-2-amine was activated with 1,1′-carbonyldiimidazole (CDI; Scheme 1a). Subsequently, intermediate **1** was reacted with corresponding substituted aniline (resp. 5-aminopyridin-2-ol for final product **8**) to give final di-substituted ureas (**2**–**11**). To obtain compounds **7**, **9** and **10**, *N*-Boc protective group was cleaved under acidic conditions (Scheme 1c) as the final step of their synthesis.

Most aniline derivatives were commercially available, but in several cases the aniline intermediates had to be prepared as further described:

In general, reduction of substituted nitrobenzenes into the corresponding anilines (e.g., **12**) was achieved with palladium on activated carbon (Pd/C) catalysed hydrogenation (Scheme 2). 

2-(*Tert*-butyl)phenol was selected as a starting material for introduction of *tert*-butyl group into the *meta* position of distal phenyl ring. Firstly, nitration was achieved with nitric acid in the presence of acetic acid as reaction solvent (Scheme 3a) to obtain intermediate **13**. Secondly, the introduced nitro group was reduced to 4-amino-2-(*tert*-butyl)phenol (**14**). Initially, the reduction was attempted with Pd/C catalysed hydrogenation (Scheme 2). However, a complex mixture of decomposed starting material was received. Thus, reduction was accomplished using iron powder and ammonium chloride (Scheme 3b) to successfully obtain intermediate **14**.

5-amino-2-hydroxybenzonitrile (**15**) was prepared from its methoxy analogue by demethylation using aluminium chloride (Scheme 4).

*N*-Boc (de)protection had to be performed in order to obtain final compounds with the free amine group on the distal phenyl ring. Firstly, the amine group of nitroaniline was protected with di-*tert*-butyl dicarbonate (Scheme 5a) to yield intermediates **16**–**18**. Secondly, the nitro group was reduced with Pd/C catalysed hydrogenation to obtain intermediates **19**–**21** (Scheme 5b). Final *N*-Boc acidic deprotection was performed after the urea formation step (Scheme 1). 

Aniline analogue with primary alcohol group in the *para* position (**22**) was generated via reduction of corresponding carboxylic acid with lithium aluminium hydride (Scheme 6).

The next series was focused on selected modifications in the linker region of the scaffold, while the original distal phenyl ring substitution (3-chlorine-4-hydroxy) was selected in combination with either 6-methoxy, 6-chlorine or unsubstituted benzothiazole ring (Table 2). Additionally, to compliment recently published work [8,17], dimethyl phosphonate analogues were prepared as standards (**34**–**36**) for comparison between inter-workgroup biological evaluations along with the most promising 3-chloro, 4-hydroxy substitution pattern. Finally, methylation of either one or both nitrogen atoms of the urea linker was conducted with the aim of constraining the conjugation between the two aromatic moieties.

Compound **23** with linker consisting of a secondary amine group was prepared by means of simple *N*-alkylation, and amides **24** and **25** were prepared in a reaction of corresponding carboxylic acid with CDI and corresponding amine. Compound **25** was subsequently *O*-demethylated to give the final product **26** (Scheme 7).

Compounds **28** and **29** were prepared using the general procedure for urea linker synthesis in reaction with CDI (Scheme 8). In case of compound **28** synthesis, the corresponding benzylamine intermediate (**27**) was first prepared from its methoxy analogue by demethylation using AlCl_3_ (Scheme 8).

While the originally used reaction conditions proved to be troublesome to produce the desired compounds [17], dimethyl phosphonates (**34**–**37**) were instead prepared in a two-step process. Firstly, 6-methoxybenzo[*d*]thiazol-2-amine and corresponding aldehyde were coupled at reflux conditions to obtain imines **30**–**33**, which were subsequently treated with dimethyl phosphite and 1,1,3,3-tetramethylguanidine to generate the final products in satisfactory yields (Scheme 9).

Compound **41** was prepared in four steps (Scheme 10). The benzothiazole moiety (**38**) was prepared from 2-iodoaniline in reaction with methylisothiocynate and tetrabutylammonium bromide catalysed by copper (I) chloride [18]. 3-chloro-4-methoxyaniline was treated with triphosgene to give the isocyanate intermediate (**39**), which was then reacted with the benzothiazole moiety and the resulting methoxy derivative (**40**) was demethylated using AlCl_3_ to give compound **41**.

The first step in the synthesis of products **45**, **46** and **49** was to prepare corresponding *N*-methylated phenyl moieties in one (*N*-methylation with methyl iodide) or actually two steps (*O*-demethylation using AlCl_3_) as shown in Scheme 11.

2-chloro-4-(methylamino)phenol (**43**) was then treated with the corresponding intermediate (**44** and **1**) produced in the reaction of benzothiazole-2-amine and 6-methoxybenzothiazole-2-amine with CDI (Scheme 12a) to give final products **45** and **46**.

Synthesis of final product **49** started with preparation of 3-(benzo[*d*]thiazol-2-yl)-1-(3-chloro-4-methoxyphenyl)-1-methylurea (**47**) from the two previously generated intermediates **42** and **44** (Scheme 12). Compound **47** was then treated with methyl iodide to methylate the other nitrogen on the urea linker (**48**). Finally, *O*-demethylation using AlCl_3_ gave the desired product **49** (Scheme 12).

The third series of compounds (Table 3) focused on evaluating substitutions within the benzothiazole ring, predominantly to exploit position 6, a key area highlighted previously [9]. Our previous findings indicated that a 6-trifluromethoxy moiety and a 6-halogen moiety, led to an increased inhibitory ability towards 17β-HSD10.

If not commercially available, the 6-substituted benzothiazole-2-amines were prepared from the corresponding 4-substituted anilines in reaction with potassium isocyanate and bromine (**50**, **51**) or potassium isocyanate and tetramethylammonium dichloroiodate (**52**). 6-thiocyanatobenzothiazol-2-amine (**53**) was obtained as a by-product during preparation of 6-iodobenzo[*d*]thiazol-2-amine (Scheme 13). The synthesis proceeded according to the general procedure using CDI to give intermediates (**54**–**60**) and final products **61**–**67** (Scheme 13).

In the fourth series the benzothiazole heterocycle itself became the subject of modifications (as indicated in Table 4). The benzene ring was replaced with a saturated cyclohexane (**71**), separated (**73**) or completely removed (**74**), and the thiazole ring was replaced with an aliphatic cyclopentane (**72**) or replaced with an ethylene bridge (**75**). Further, the whole benzothiazole moiety was flipped and attached to urea via carbon in position 6 of the heterocycle. Moreover, the symmetric derivative (**78**) was prepared to find out whether the dimerized phenyl moiety alone is sufficient for 17β-HSD10 inhibition.

The general procedure for synthesis of the urea molecules described earlier in the text (Scheme 1) was only suitable for compounds comprising the 2-aminothiazole core (imidazolecarboxamide intermediates **68**–**70** and final products **71**, **73** and **74**). Therefore, for compounds **72**, **75** and **76**, the synthesis procedure had to be updated due to an increase in the solubility of imidazolecarboxamide intermediates, which did not allow for their simple isolation by filtration in satisfactory yields. Consequently, after the activation of starting compound with CDI was completed, 4-amino-2-chlorophenol was added directly to the current reaction mixture (Scheme 14).

The symmetric 1,3-bis(3-chloro-4-hydroxyphenyl)urea (**78**) was prepared in two steps (Scheme 15). First, 3-chloro-4-methoxyaniline was treated with CDI to give 1,3-bis(3-chloro-4-methoxyphenyl)urea (**77**), which was then *O*-demethylated in reaction with AlCl_3_.

### 2.2. Biochemical and Biophysical Evaluation

In order to reduce attrition rates and improve assay reproducibility we have developed a high throughput screening (HTS) pipeline (Figure 3 [19]). In brief, compounds are screened in the recombinant 17β-HSD10 enzyme activity assay (Table 5, Figure 4). Our best previously published compounds have set the threshold of 40% remaining 17β-HSD10 activity as a minimum standard [9] and if compounds can better this threshold, they are further screened using our orthogonal counter assays, dose response assays and kinetic assessment (Table 5). Finally, if passing these criteria with favourable characteristics, the compounds progress into cellular evaluation through cytotoxicity testing and measuring 17β-HSD10 activity within cells (Table 6). 

### 2.3. Primary Enzyme Assay Results 

Full results for the primary nicotinamide adenine dinucleotide (NADH) assay screens are shown in Figure 4 including our four best previously published compounds for comparison (Figure 1) [9]. 

Our first analogue series (**2**–**11**; Table 1 and Figure 4) focused on establishing how alterations to the 3 and 4 position on the distal (phenolic) ring affect inhibition potency. In this series, compounds **5** and **6** showed a huge improvement in potency with remaining 17β-HSD10 activity of 13.45% and 6.72%, respectively, at 25 μM. A significant finding from our previous work indicated that a *p*-hydroxy along *m*-chlorine substitution pattern displayed the most pronounced inhibitory activity [9], and a deviation from the 3-halogen and 4-hydroxyl pattern resulted in a dramatic decrease in 17β-HSD10 inhibition. Our findings in this series further support this, and establish that the bulkier, 3-bromo and 3-iodo substitutions are even more favourable at this position. Replacement of the phenolic hydroxyl with an amine or methylhydroxy group led to loss of activity, which further confirms the importance of the 4-positioned phenolic hydroxyl as was previously suggested [9].

The second analogue series (**23**–**49**; Table 2) focused on evaluating changes to the urea linker. Unfortunately, any variation from the urea linker resulted in a dramatic decrease of inhibitory activity (Figure 4). This was further supported by the inclusion of compounds (**34**–**36**) previously published by Valasani et al. with the inclusion of our novel phosphonate compound (**37**) determining that a phosphonate linker did not increase 17β-HSD10 inhibition. Indeed, all linker variations resulted in negligible changes to 17β-HSD10 activity with the exception of compound **24**. Although the secondary amide substitution (with amide nitrogen attached to benzothiazole moiety) in compound **24** appeared slightly more favourable than most, it was still not as potent as the original urea moiety and just outside of the threshold for further analysis at 47.19% 17β-HSD10 activity remaining at 25 μM (Figure 4). Mono and dimethylation of the urea linker to enforce sp^3^, rather than sp^2^ character, showed a clear detrimental effect on the activity in compound **45**–**49**.

The third series of compounds (**61**–**67**) focused on evaluating substitutions within the benzothiazole ring, predominantly to exploit position 6, a key area highlighted previously (Hroch et al. 2016). Our previous findings indicated that a 6-trifluromethoxy moiety and a 6-halogen moiety led to an increased inhibitory ability towards 17β-HSD10. This series appear to be the most promising displaying the largest decrease in 17β-HSD10 activity (indicated in Figure 4), in particular when bulky substitutions at position 6 were applied. This is particularly apparent in compounds **61** and **62** whereby, as the functional group size increases at position 6, 17β-HSD10 activity decreases with 6-*t*-butyl (**62**) inhibiting 17β-HSD10 by 78.36% and the 6-isopropyl substitution (**61**) inhibiting 17β-HSD10 by 77.37% at 25 μM. Other key structure activity relationships in this series have been established through 6-thiol additions (**64**–**67**). Again, bulkier substituents produce a larger inhibitory effect with the 6-sulfonyl (**66)** less effective than the 6-trifluoromethylsulfonyl (**67**) with 32.16% and 16.16% 17β-HSD10 activity remaining at 25 μM. The most potent compound in this series (**65**) introduced the pseudo-halogenic 6-thiocyanate moiety, which inhibited 17β-HSD10 activity by 85.69% at 25 μM. 

In order to validate the importance of the benzothiazol-2-yl moiety, several structural analogues were prepared and evaluated in the fourth series (**71**–**78**; Table 4). Indeed, any deviation from the benzothiazol-2-yl moiety resulted in decreased biological activity and only the symmetrical compound (**78**) showed real inhibitory activity with 17β-HSD10 activity reduced to 40.69% at 25 μM (Figure 4).

Overall, after primary screening we were left with 10 compounds which would progress down the pipeline: compounds **5**, **6**, **61**, **62**, **63**, **64**, **65**, **66**, **67** and **78**.

### 2.4. Orthogonal Counter Screens

During our enzymatic assay development, it was noted that the assay was susceptible to false positives through redox cycling and aggregation mechanism [19], therefore, two orthogonal counter screens have been implemented to validate the primary screen results. The addition of the detergent Triton X-100 to the assay buffer prevents the hydrophobic interactions required for aggregation, by which a reduction in inhibition in the presence of Triton X-100 indicates the undesirable inhibitory mode of action whereby, the compound could be potentially inhibiting the enzyme through the indirect sequestration of the protein. Results (Table 5) identify compound **63** as a potential aggregator as it showed a 61% increase in 17β-HSD10 activity in the presence of Triton-X100. Compounds **65** and **67** also showed some reduction in activity but this is much less pronounced. Given that these three compounds were part of a small series we decided to advance them into the next step of screening.

With the inclusion of the strong reducing agent dithiothreitol (DTT) in the assay buffer, compounds can appear as a false positive as DTT is capable of generating H_2_O_2_ causing indirect enzyme inhibition and assay interference. The fluorescence change during the reduction of resazurin to resorufin can be measured as an indication of any redox cycling compounds. The results indicate that none of the compounds appear to be acting via this undesirable mode of action (Table 5).

### 2.5. Dose Response and Kinetic Evaluation

Our most promising compounds demonstrate reasonable IC_50_ values of around 1–2 μM (Table 5 and graphs in Appendix A). Significantly, these compounds all display a mixed mechanism of inhibition with respect to both substrate acetoacetyl-Coenzyme A and co-factor NADH whereby at low concentrations they appear to act in a competitive manner, but at high concentrations they are inhibiting in other sites (Table 5, Hanes–Woolf plots in Appendix A). This is favourable over other previously published work [7,20] as the AG18051 compound irreversibly inhibits 17β-HSD10, forming a covalent adduct with NADH at the active site, thus introducing a potential specificity issue. 

### 2.6. Cellular Screening

Compound toxicity and potency was also assessed using HEK293 mts17β-HSD10 cells; results are shown in Table 6. Our fluorogenic probe, (−)-CHANA, a 17β-HSD10 substrate [21] was used to calculate cellular IC_50_ values (graphs in Appendix A) with the exception of compounds **64**, **66**, **67** and **78**, which were precipitating within the assay media and not able to effectively penetrate into cells. Compounds **61**, **62** and **63** proved to be the most potent in our cellular assay with IC_50_ values of 7.88, 3.77 and 2.29 μM, respectively. 

Lactate dehydrogenase (LDH) is a colorimetric assay routinely used to quantitatively measure LDH released into the media from damaged cells as a biomarker for cellular cytotoxicity and cytolysis. HEK293 mts17β-HSD10 cells were treated with compound (100 and 25 μM**)** for 24 h before measurements were taken. At 25 μM compounds showed around 10–30% cytotoxicity, however, this concentration is substantially higher than the measured IC_50_ values and as such is not a cause for concern. Compound **65** showed a remarkably higher IC_50_ value in the (−)-CHANA assay which suggests that the uptake of compound by cells and 17β-HSD10 target engagement is not as favourable as others.

## 3. Materials and Methods

### 3.1. General Chemistry

All reagents and solvents were purchased from commercial sources (Sigma Aldrich, Prague, Czech Republic; Activate Scientific, Prien, Germany; Alfa Aesar, Kandel, Germany; Merck, Darmstadt, Germany; Penta Chemicals, Prague, Czech Republic and VWR, Stribrna Skalice, Czech Republic) and they were used without any further purification. Low boiling point (≥90% 40–60 °C) petroleum ether (PE) was used if not stated otherwise. 

Thin-layer chromatography (TLC) for reaction monitoring was performed on Merck aluminium sheets, silica gel 60 F_254_ (Darmstadt, Germany). Visualisation was performed either via UV (254 nm) or appropriate stain reagent solutions (alternatively in combination of both). Preparative column chromatography was performed on silica gel 60 (70–230 mesh, 63–200 μm, 60 Å pore size). Melting points were determined on a Stuart SMP30 melting point apparatus and are uncorrected. 

Nuclear magnetic resonance (NMR) spectra were acquired at 500/126/202 MHz (^1^H, ^13^C and ^31^P) on a Varian S500 spectrometer or at 300/75 MHz (^1^H and ^13^C) on a Varian Gemini 300 spectrometer (both produced by Palo Alto, CA, USA). Chemical shifts δ are given in ppm and referenced to the signal center of solvent peaks (DMSO-*d*_6_: δ 2.50 ppm and 39.52 ppm for ^1^H and ^13^C, respectively; Chloroform-*d*: δ 7.26 ppm and 77.16 ppm for ^1^H and ^13^C, respectively), thus indirectly correlated to TMS standard (δ 0 ppm). Chemical shifts δ for ^31^P are given in ppm and referenced to the phosphoric acid standard (δ 0 ppm). Coupling constants are expressed in Hz.

High-resolution mass spectra (HRMS) were recorded by coupled LC-MS system consisting of Dionex UltiMate 3000 analytical LC system and Q Exactive Plus hybrid quadrupole-orbitrap spectrometer (both produced by ThermoFisher Scientific, Bremen, Germany). As an ion-source, heated electro-spray ionization (HESI) was utilised (setting: sheath gas flow rate 40, aux gas flow rate 10, sweep gas flow rate 2, spray voltage 3.2 kV, capillary temperature 350 °C, aux gas temperature 300 °C, S-lens RF level 50). Positive ions were monitored in the range of 100–1500 *m/z* with the resolution set to 140,000. Obtained mass spectra were processed in Xcalibur 3.0.63 software (ThermoFisher Scientific, Bremen, Germany).

Further synthetic information can be found in the Appendix A. 

### 3.2. Final Products Characterization

The purification method is specified here only when altered from the generally used method described in Appendix A.

*1-(4-Hydroxy-3-methylphenyl)-3-(6-methoxybenzo[d]thiazol-2-yl)urea* (**2**) Yield 85%; mp: 262–263 °C; ^1^H-NMR (500 MHz, DMSO-*d*_6_): δ (ppm) 10.55 (br s, 1H), 9.09 (s, 1H), 8.77 (s, 1H), 7.54 (d, *J* = 8.8 Hz, 1H), 7.50 (d, *J* = 2.6 Hz, 1H), 7.18 (d, *J* = 2.6 Hz, 1H), 7.10 (dd, *J* = 8.5, 2.7 Hz, 1H), 6.97 (dd, *J* = 8.8, 2.6 Hz, 1H), 6.73 (d, *J* = 8.5 Hz, 1H), 3.79 (s, 3H), 2.12 (s, 3H); ^13^C-NMR (126 MHz, DMSO-*d*_6_): δ (ppm) 157.57, 155.61, 151.75, 151.47, 142.84, 132.53, 129.56, 124.11, 122.24, 120.13, 118.18, 114.64, 114.29, 104.87, 55.60, 16.12; HRMS (ESI) calcd for C_16_H_16_N_3_O_3_S [M + H]^+^ 330.09069, found 330.09039. 

*1-(3-(Tert-butyl)-4-hydroxyphenyl)-3-(6-methoxybenzo[d]thiazol-2-yl)urea* (**3**) Yield 73%; mp: 259–260 °C; ^1^H-NMR (500 MHz, DMSO-*d*_6_): δ (ppm) 10.47 (br s, 1H), 9.18 (br s, 1H), 8.79 (s, 1H), 7.54 (d, *J* = 8.8 Hz, 1H), 7.49 (d, *J* = 2.6 Hz, 1H), 7.19 (d, *J* = 2.6 Hz, 1H), 7.17 (dd, *J* = 8.4, 2.6 Hz, 1H), 6.97 (dd, *J* = 8.8, 2.6 Hz, 1H), 6.73 (d, *J* = 8.4 Hz, 1H), 3.79 (s, 3H), 1.35 (s, 9H); ^13^C-NMR (126 MHz, DMSO-*d*_6_): δ (ppm) 157.50, 155.61, 151.91, 151.84, 142.82, 135.53, 132.60, 129.43, 120.20, 118.58, 118.38, 116.13, 114.28, 104.90, 55.62, 34.35, 29.25; HRMS (ESI) calcd for C_19_H_22_N_3_O_3_S [M + H]^+^ 372.13764, found 372.13730. 

*1-(3-Cyano-4-hydroxyphenyl)-3-(6-methoxybenzo[d]thiazol-2-yl)urea* (**4**) Yield 98%; mp: 277–279 °C; ^1^H-NMR (500 MHz, DMSO-*d*_6_): δ (ppm) 10.86 (s, 1H), 9.37 (s, 1H), 7.76 (d, *J* = 2.7 Hz, 1H), 7.56 – 7.52 (m, 2H), 7.51 (d, *J* = 2.6 Hz, 1H), 7.01 (d, *J* = 9.0 Hz, 1H), 6.98 (dd, *J* = 8.8, 2.6 Hz, 1H), 3.79 (s, 3H); ^13^C-NMR (126 MHz, DMSO-*d*_6_): δ (ppm) 157.84, 156.05, 155.71, 152.44, 141.78, 132.24, 130.54, 126.52, 122.87, 119.79, 116.80, 116.73, 114.40, 105.00, 98.49, 55.61; HRMS (ESI) calcd for C_16_H_12_N_4_O_3_S [M + H]^+^ 341.07029, found 341.07016.

*1-(3-Bromo-4-hydroxyphenyl)-3-(6-methoxybenzo[d]thiazol-2-yl)urea* (**5**) Yield 82%; mp: 246–247 °C; ^1^H-NMR (500 MHz, DMSO-*d*_6_): δ (ppm) 10.71 (br s, 1H), 9.99 (s, 1H), 8.97 (s, 1H), 7.75 (s, 1H), 7.54 (d, *J* = 8.4 Hz, 1H), 7.50 (s, 1H), 7.22 (d, *J* = 8.6 Hz, 1H), 6.97 (d, *J* = 8.5 Hz, 1H), 6.92 (d, *J* = 8.6 Hz, 1H), 3.79 (s, 3H); ^13^C-NMR (126 MHz, DMSO-*d*_6_): δ (ppm) 157.84, 155.67, 152.21, 149.97, 142.25, 132.33, 131.04, 123.65, 120.14, 119.87, 116.28, 114.37, 108.86, 104.95, 55.61; HRMS (ESI) calcd for C_15_H_13_BrN_3_O_3_S [M + H]^+^ 393.98555, found 393.98489. 

*1-(4-Hydroxy-3-iodophenyl)-3-(6-methoxybenzo[d]thiazol-2-yl)urea* (**6**) Yield 85%; mp: 241–242 °C; ^1^H-NMR (300 MHz, DMSO-*d*_6_): δ (ppm) 10.07 (br s, 1H), 9.05 (s, 1H), 7.90 (d, *J* = 2.6 Hz, 1H), 7.54 (d, *J* = 8.8 Hz, 1H), 7.51 (d, *J* = 2.6 Hz, 1H), 7.25 (dd, *J* = 8.7, 2.6 Hz, 1H), 6.97 (dd, *J* = 8.8, 2.6 Hz, 1H), 6.85 (d, *J* = 8.7 Hz, 1H), 3.79 (s, 3H); ^13^C-NMR (75 MHz, DMSO-*d*_6_): δ (ppm) 157.72, 155.67, 152.67, 152.10, 142.15, 132.35, 131.29, 129.42, 120.99, 119.93, 114.75, 114.38, 104.93, 84.10, 55.62; HRMS (ESI) calcd for C_15_H_13_IN_3_O_3_S [M + H]^+^ 441.97168, found 441.97049. 

*1-(3-Amino-4-hydroxyphenyl)-3-(6-methoxybenzo[d]thiazol-2-yl)urea* (**7**) Yield 53%; mp: 180–181 °C; ^1^H-NMR (500 MHz, DMSO-*d*_6_): δ (ppm) 10.41 (br s, 1H), 8.76 (br s, 1H), 8.65 (s, 1H), 7.53 (d, *J* = 8.8 Hz, 1H), 7.49 (d, *J* = 2.6 Hz, 1H), 6.96 (dd, *J* = 8.8, 2.6 Hz, 1H), 6.82 (d, *J* = 2.6 Hz, 1H), 6.57 (d, *J* = 8.3 Hz, 1H), 6.47 (dd, *J* = 8.3, 2.5 Hz, 1H), 4.61 (s, 2H), 3.79 (s, 3H); ^13^C-NMR (126 MHz, DMSO-*d*_6_): δ (ppm) 157.45, 155.57, 151.34, 143.00, 140.11, 136.90, 132.57, 130.42, 122.10, 120.22, 114.24, 107.26, 106.17, 104.84, 55.58; HRMS (ESI) calcd for C_15_H_15_N_4_O_3_S [M + H]^+^ 331.08594, found 331.08527. 

*1-(6-Hydroxypyridin-3-yl)-3-(6-methoxybenzo[d]thiazol-2-yl)urea* (**8**) Yield 82%; mp: 272–273 °C; ^1^H-NMR (300 MHz, DMSO-*d*_6_): δ (ppm) 11.38 (br s, 1H), 9.08 (br s, 1H), 7.64 (s, 1H), 7.59 (d, *J* = 9.1 Hz, 1H), 7.53 – 7.39 (m, 2H), 6.97 (dd, *J* = 9.2, 2.6 Hz, 1H), 6.36 (d, *J* = 10.1 Hz, 1H), 3.79 (s, 3H); ^13^C-NMR (75 MHz, DMSO-*d*_6_): δ (ppm) 160.85, 158.50, 155.83, 153.21, 142.19, 138.04, 132.42, 127.92, 120.11, 119.28, 119.17, 114.49, 105.11, 55.78; HRMS (ESI) calcd for C_14_H_13_N_4_O_3_S [M + H]^+^ 317.07029, found 317.07004. 

*1-(4-Aminophenyl)-3-(6-methoxybenzo[d]thiazol-2-yl)urea* (**9**) Yield 93%; mp: 304–306 °C (decomp); ^1^H-NMR (500 MHz, DMSO-*d*_6_): δ (ppm) 10.46 (br s, 1H), 8.63 (s, 1H), 7.54 (d, *J* = 8.8 Hz, 1H), 7.50 (d, *J* = 2.6 Hz, 1H), 7.15 – 7.10 (m, 2H), 6.96 (dd, *J* = 8.8, 2.6 Hz, 1H), 6.57 – 6.52 (m, 2H), 4.90 (s, 2H), 3.79 (s, 3H); ^13^C-NMR (126 MHz, DMSO-*d*_6_): δ (ppm) 157.58, 155.56, 151.75, 144.96, 142.74, 132.56, 127.01, 121.23, 120.12, 114.23, 114.05, 104.87, 55.58; HRMS (ESI) calcd for C_15_H_15_N_4_O_2_S [M + H]^+^ 315.09102, found 315.09058. 

*1-(4-Amino-3-chlorophenyl)-3-(6-methoxybenzo[d]thiazol-2-yl)urea* (**10**) Yield 78%y mp: 310-311 °C (decomp); ^1^H-NMR (500 MHz, DMSO-*d*_6_): δ (ppm) 10.60 (br s, 1H), 8.82 (s, 1H), 7.54 (d, *J* = 8.7 Hz, 1H), 7.50 (d, *J* = 2.6 Hz, 1H), 7.47 (d, *J* = 2.4 Hz, 1H), 7.06 (dd, *J* = 8.6, 2.4 Hz, 1H), 6.97 (dd, *J* = 8.8, 2.6 Hz, 1H), 6.78 (d, *J* = 8.6 Hz, 1H), 5.14 (s, 2H), 3.79 (s, 3H); ^13^C-NMR (126 MHz, DMSO-*d*_6_): δ (ppm) 157.69, 155.61, 151.90, 140.78, 132.42, 128.05, 120.50, 120.23, 119.97, 116.77, 115.48, 114.29, 104.91, 55.59; HRMS (ESI) calcd for C_15_H_14_ClN_4_O_2_S [M + H]^+^ 349.05205, found 349.05154. 

*1-(3-Chloro-4-(hydroxymethyl)phenyl)-3-(6-methoxybenzo[d]thiazol-2-yl)urea (**11**)* Yield 92%; mp: 229–230 °C; ^1^H-NMR (500 MHz, DMSO-*d*_6_): δ (ppm) 10.84 (br s, 1H), 9.26 (s, 1H), 7.72 (d, *J* = 2.2 Hz, 1H), 7.55 (d, *J* = 8.7 Hz, 1H), 7.51 (d, *J* = 2.6 Hz, 1H), 7.47 (d, *J* = 8.4 Hz, 1H), 7.38 (dd, *J* = 8.4 Hz, 2.1 Hz, 1H), 6.98 (dd, *J* = 8.8, 2.6 Hz, 1H), 5.29 (t, *J* = 5.6 Hz, 1H), 4.53 (d, *J* = 5.2 Hz, 2H), 3.80 (s, 3H); ^13^C-NMR (126 MHz, DMSO-*d*_6_): δ (ppm) 157.75, 155.74, 152.27, 138.45, 133.63, 132.18, 131.19, 128.67, 127.18, 119.85, 118.56, 117.35, 114.44, 105.01, 59.99, 55.61; HRMS (ESI) calcd for C_16_H_15_ClN_3_O_3_S [M + H]^+^ 364.05172, found 364.05103.

*2-Chloro-4-((6-chlorobenzo[d]thiazol-2-yl)amino)phenol* (**23**) Yield 30%; mp: 213–214.5 °C; ^1^H-NMR (500 MHz, DMSO-*d*_6_): δ (ppm) 10.49 (br s, 1H), 9.92 (br s, 1H), 7.94 – 7.85 (m, 2H), 7.54 (d, *J* = 8.6 Hz, 1H), 7.40 (dd, *J* = 8.8, 2.6 Hz, 1H), 7.31 (dd, *J* = 8.6, 2.2 Hz, 1H), 6.98 (d, *J* = 8.8 Hz, 1H); ^13^C-NMR (126 MHz, DMSO-*d*_6_): δ (ppm) 162.53, 150.83, 148.56, 132.94, 131.53, 126.01, 125.88, 120.74, 119.86, 119.74, 119.43, 118.46, 116.87; HRMS (ESI) calcd for C_13_H_8_Cl_2_N_2_OS [M + H]^+^ 310.98072, found 310.98016.

*3-Chloro-4-hydroxy-N-(6-methoxybenzo[d]thiazol-2-yl)benzamide* (**24**) Yield 69%; mp: 301.5–302.5 °C; ^1^H-NMR (300 MHz, DMSO-*d*_6_): δ (ppm) 12.51 (br s, 1H), 11.25 (br s, 1H), 8.21 (s, 1H), 7.97 (d, *J* = 8.6 Hz, 1H), 7.66 (d, *J* = 8.9 Hz, 1H), 7.59 (s, 1H), 7.17 – 6.96 (m, 2H), 3.82 (s, 3H); ^13^C-NMR (75 MHz, DMSO-*d*_6_): δ (ppm) 164.08, 157.27, 156.85, 156.21, 142.57, 132.85, 130.34, 128.93, 123.50, 120.96, 119.89, 116.34, 115.00, 104.66, 55.64; HRMS (ESI) calcd for C_15_H_11_ClN_2_O_3_S [M + H]^+^ 335.02517, found 335.02466.

*N-(3-Chloro-4-hydroxyphenyl)-6-methoxybenzo[d]thiazole-2-carboxamide* (**26**) Yield 58%; mp: 260–261.5 °C; ^1^H-NMR (500 MHz, DMSO-*d*_6_): δ (ppm) 10.94 (br s, 1H), 10.09 (br s, 1H), 8.06 (d, *J* = 8.4 Hz, 1H), 7.94 (s, 1H), 7.80 (s, 1H), 7.65 (d, *J* = 7.7 Hz, 1H), 7.23 (d, *J* = 8.1 Hz, 1H), 6.97 (d, *J* = 8.2 Hz, 1H), 3.88 (s, 3H); ^13^C-NMR (126 MHz, DMSO-*d*_6_): δ (ppm) 161.76, 158.62, 157.88, 149.87, 147.05, 138.22, 130.34, 124.71, 122.22, 120.81, 119.09, 117.30, 116.36, 104.79, 55.84; HRMS (ESI) calcd for C_15_H_11_ClN_2_O_3_S [M + H]^+^ 335.02517, found 335.02466.

*1-(3-Chloro-4-hydroxybenzyl)-3-(6-methoxybenzo[d]thiazol-2-yl)urea* (**28**)

The crude product was purified using column chromatography.

Yield 20%; mp: 249–251 °C; ^1^H-NMR (500 MHz, DMSO-*d*_6_): δ (ppm) 10.60 (br s, 1H), 10.08 (br s, 1H), 7.51 (d, *J* = 8.8 Hz, 1H), 7.48 (d, *J* = 2.6 Hz, 1H), 7.29 (d, *J* = 2.1 Hz, 1H), 7.13 (t, *J* = 5.3 Hz, 1H), 7.10 (dd, *J* = 8.3, 2.1 Hz, 1H), 6.98 – 6.90 (m, 2H), 4.25 (d, *J* = 5.9 Hz, 2H), 3.78 (s, 3H); ^13^C-NMR (126 MHz, DMSO-*d*_6_): δ (ppm) 157.80, 155.52, 153.84, 152.01, 143.16, 132.59, 131.25, 128.80, 127.18, 120.19, 119.37, 116.54, 114.15, 104.81, 55.57, 42.01; HRMS (ESI) calcd for C_16_H_14_ClN_3_O_3_S [M + H]^+^ 364.05172, found 364.05115.

*1-(3,4-Dihydroxybenzyl)-3-(6-methoxybenzo[d]thiazol-2-yl)urea* (**29**)

After the reaction was completed (monitored by TLC), 1M aq. HCl was poured to the reaction mixture and the product was extracted to DCM. The organic layer was concentrated and the crude product was recrystallized from MeCN.

Yield 50%; mp: 141.5–142 °C; ^1^H-NMR (500 MHz, DMSO-*d*_6_): δ (ppm) 7.51 (d, *J* = 8.8 Hz, 1H), 7.48 (d, *J* = 2.6 Hz, 1H), 7.15 (br s, 1H), 6.95 (dd, *J* = 8.8, 2.6 Hz, 1H), 6.71 (d, *J* = 2.0 Hz, 1H), 6.68 (d, *J* = 8.0 Hz, 1H), 6.56 (dd, *J* = 8.0, 2.0 Hz, 1H), 4.18 (d, *J* = 5.5 Hz, 2H), 3.78 (s, 3H); ^13^C-NMR (126 MHz, DMSO-*d*_6_): δ (ppm) 158.51, 155.74, 153.64, 145.23, 144.41, 141.36, 131.95, 129.93, 119.61, 118.26, 115.49, 114.93, 114.45, 105.07, 55.65, 42.68; HRMS (ESI) calcd for C_16_H_15_N_3_O_4_S [M + H]^+^ 346.08560, found 346.08517.

Dimethyl ((4-fluorophenyl)((6-methoxybenzo[d]thiazol-2-yl)amino)methyl) phosphonate (**34**) (Valasani et al. 2013). 

Yield 65%; mp: 173–174 °C; ^1^H-NMR (500 MHz, DMSO-*d*_6_): δ (ppm) 9.48 (s, 1H), 8.76 (dd, *J* = 9.7, 2.9 Hz, 1H), 7.33 – 7.27 (m, 4H), 6.82 (dd, *J* = 8.7, 2.7 Hz, 1H), 6.75 (d, *J* = 8.4 Hz, 2H), 5.54 (dd, *J* = 20.9, 9.7 Hz, 1H), 3.72 (s, 3H), 3.64 (d, *J* = 10.5 Hz, 3H), 3.50 (d, *J* = 10.5 Hz, 3H); ^13^C-NMR (126 MHz, DMSO-*d*_6_): δ (ppm) 163.62 (d, *J* = 10.0 Hz), 161.69 (dd, *J* = 244.1, 2.9 Hz), 154.67, 145.52, 132.10 (d, *J* = 2.9 Hz), 131.87, 130.02 (dd, *J* = 8.3, 5.5 Hz), 118.79, 115.18 (dd, *J* = 21.6, 1.7 Hz), 113.09, 105.58, 53.53 (d, *J* = 6.8 Hz), 53.25 (d, *J* = 6.8 Hz), 53.12 (d, *J* = 154.1 Hz); ^31^P NMR (202 MHz, DMSO-*d*_6_): δ (ppm) 23.56 (d, *J* = 4.5 Hz); HRMS (ESI) calcd for C_17_H_19_FN_2_O_4_PS [M + H]^+^ 397.07817, found 397.07755. 

*Dimethyl ((4-hydroxyphenyl)((6-methoxybenzo[d]thiazol-2-yl)amino)methyl) phosphonate* (**35**) (Valasani et al. 2013). Yield 76%; mp: 217–218 °C; ^1^H-NMR (500 MHz, DMSO-*d*_6_): δ (ppm) 9.48 (s, 1H), 8.76 (dd, *J* = 9.7, 2.9 Hz, 1H), 7.33 – 7.28 (m, 4H), 6.82 (dd, *J* = 8.7, 2.7 Hz, 1H), 6.75 (d, *J* = 8.4 Hz, 2H), 5.54 (dd, *J* = 20.9, 9.7 Hz, 1H), 3.72 (s, 3H), 3.64 (d, *J* = 10.5 Hz, 3H), 3.50 (d, *J* = 10.5 Hz, 3H); ^13^C-NMR (126 MHz, DMSO-*d*_6_): δ (ppm) 163.70 (d, *J* = 10.0 Hz), 157.06 (d, *J* = 2.4 Hz), 154.57, 145.64, 131.82, 129.31 (d, *J* = 5.7 Hz), 125.77, 118.68, 115.08, 113.01, 105.57, 55.53, 53.33 (d, *J* = 6.9 Hz), 53.28 (d, *J* = 155.2 Hz), 53.14 (d, *J* = 6.9 Hz); ^31^P NMR (202 MHz, DMSO-*d*_6_): δ (ppm) 24.25; HRMS (ESI) calcd for C_17_H_20_N_2_O_5_PS [M + H]^+^ 395.08251, found 395.08215. 

*Methyl 5-((dimethoxyphosphoryl)((6-methoxybenzo[d]thiazol-2-yl)amino)methyl)-2- hydroxybenzoate* (**36**) (Valasani et al. 2013) Yield 73%; mp: 180–181 °C; ^1^H-NMR (500 MHz, DMSO-*d*_6_): δ (ppm) 10.53 (s, 1H), 8.90 (dd, *J* = 9.6, 3.5 Hz, 1H), 7.95 (t, *J* = 2.3 Hz, 1H), 7.65 (dt, *J* = 8.7, 2.1 Hz, 1H), 7.32 (d, *J* = 2.7 Hz, 1H), 7.30 (d, *J* = 8.7 Hz, 1H), 7.01 (d, *J* = 8.6 Hz, 1H), 6.82 (dd, *J* = 8.8, 2.7 Hz, 1H), 5.63 (dd, *J* = 21.3, 9.4 Hz, 1H), 3.91 (s, 3H), 3.72 (s, 3H), 3.67 (d, *J* = 10.6 Hz, 3H), 3.55 (d, *J* = 10.6 Hz, 3H); ^13^C-NMR (126 MHz, DMSO-*d*_6_): δ (ppm) 168.86, 163.61 (d, *J* = 10.1 Hz), 159.53 (d, *J* = 2.0 Hz), 154.65, 145.52, 135.35 (d, *J* = 5.1 Hz), 131.88, 129.33 (d, *J* = 6.0 Hz), 126.83, 118.79, 117.49, 105.58, 113.07, 113.03 (d, *J* = 1.9 Hz), 55.53, 53.55 (d, *J* = 7.1 Hz), 53.26 (d, *J* = 6.8 Hz), 52.90 (d, *J* = 155.2 Hz), 52.53; ^31^P NMR (202 MHz, DMSO-*d*_6_): δ (ppm) 23.67; HRMS (ESI) calcd for C_19_H_22_N_2_O_7_PS [M + H]^+^ 453.08798, found 453.08701. 

*Dimethyl ((3-chloro-4-hydroxyphenyl)((6-methoxybenzo[d]thiazol-2-yl)amino)methyl) phosphonate* (**37**) Yield 65%; mp: 198–199 °C; ^1^H-NMR (500 MHz, DMSO-*d*_6_): δ (ppm) 10.28 (s, 1H), 8.77 (dd, *J* = 9.7, 3.1 Hz, 1H), 7.50 (t, *J* = 2.1 Hz, 1H), 7.32 (d, *J* = 2.7 Hz, 1H), 7.31 (d, *J* = 8.8 Hz, 1H), 7.27 (dt, *J* = 8.5, 2.1 Hz, 1H), 6.95 (d, *J* = 8.4 Hz, 1H), 6.82 (dd, *J* = 8.7, 2.6 Hz, 1H), 5.57 (dd, *J* = 21.0, 9.6 Hz, 1H), 3.73 (s, 3H), 3.66 (d, *J* = 10.6 Hz, 3H), 3.55 (d, *J* = 10.6 Hz, 3H); ^13^C-NMR (126 MHz, DMSO-*d*_6_): δ (ppm) 163.76 (d, *J* = 10.1 Hz), 154.82, 152.90 (d, *J* = 2.4 Hz), 145.72, 132.01, 129.40 (d, *J* = 5.4 Hz), 128.12 (d, *J* = 5.9 Hz), 127.58, 119.66 (d, *J* = 2.2 Hz), 118.94, 116.52, 113.25, 105.76, 55.71, 53.66 (d, *J* = 6.8 Hz), 53.41 (d, *J* = 7.0 Hz), 52.95 (d, *J* = 155.3 Hz); ^31^P NMR (202 MHz, DMSO-*d*_6_): δ (ppm) 23.71; HRMS (ESI) calcd for C_17_H_19_ClN_2_O_5_PS [M + H]^+^ 429.04353, found 429.0425. 

*1-(Benzo[d]thiazol-2-yl)-3-(3-chloro-4-hydroxyphenyl)-1-methylurea* (**41**) Yield 83%; mp: 171–172 °C; ^1^H-NMR (500 MHz, DMSO-*d*_6_): δ (ppm) 10.02 (s, 1H), 9.47 (s, 1H), 7.94 – 7.84 (m, 1H), 7.74 (d, *J* = 7.9 Hz, 1H), 7.57 (d, *J* = 2.6 Hz, 1H), 7.45 – 7.35 (m, 1H), 7.31 (dd, *J* = 8.8, 2.6 Hz, 1H), 7.28 – 7.19 (m, 1H), 6.96 (d, *J* = 8.7 Hz, 1H), 3.75 (s, 3H); ^13^C-NMR (126 MHz, DMSO-*d*_6_): δ (ppm) 161.45, 153.45, 149.66, 148.45, 132.76, 130.50, 125.80, 123.42, 123.09, 121.95, 121.12, 120.29, 119.01, 116.29, 34.55 (d, *J* = 3.0 Hz); HRMS (ESI) calcd for C_15_H_12_ClN_3_O_2_S [M + H]^+^ 334.04115, found 334.04080.

*3-(Benzo[d]thiazol-2-yl)-1-(3-chloro-4-hydroxyphenyl)-1-methylurea* (**45**)

The crude product was dissolved in Et_2_O and filtered. To the filtrate was added PE and the solution was left to crystallize in a freezer. Filtration gave the desired pure product.

Yield 53%; mp: 139–141 °C; ^1^H-NMR (500 MHz, DMSO-*d*_6_): δ (ppm) 10.28 (br s, 1H), 7.81 (d, *J* = 7.5 Hz, 1H), 7.45 (br s, 1H), 7.38 – 7.30 (m, 2H), 7.19 (t, *J* = 7.9 Hz, 1H), 7.11 (dd, *J* = 8.6, 2.5 Hz, 1H), 6.99 (d, *J* = 8.6 Hz, 1H), 3.26 (s, 3H); ^13^C-NMR (126 MHz, DMSO-*d*_6_): δ (ppm) 151.90, 135.11, 128.70, 126.89, 125.94, 122.65, 121.62, 119.46, 116.66, 37.93; HRMS (ESI) calcd for C_15_H_12_ClN_3_O_2_S [M + H]^+^ 334.0412, found 334.0414.

*1-(3-Chloro-4-hydroxyphenyl)-3-(6-methoxybenzo[d]thiazol-2-yl)-1-methylurea* (**46**)

The crude product was purified using column chromatography.

Yield 50%; mp: 220 °C decomp.; ^1^H-NMR (500 MHz, DMSO-*d*_6_): δ (ppm) 10.28 (br s, 1H), 7.45 (d, *J* = 2.2 Hz, 1H), 7.38 (d, *J* = 8.1 Hz, 1H), 7.36 (d, *J* = 2.5 Hz, 1H), 7.11 (dd, *J* = 8.6, 2.5 Hz, 1H), 6.98 (d, *J* = 8.6 Hz, 1H), 6.94 (dd, *J* = 8.8, 2.6 Hz, 1H), 3.77 (s, 3H), 3.25 (s, 3H); ^13^C-NMR (126 MHz, DMSO-*d*_6_): δ (ppm) 155.57, 151.97, 134.94, 131.53, 128.78, 126.95, 119.52, 118.40, 116.70, 114.18, 105.12, 55.60, 37.94; HRMS (ESI) calcd for C_16_H_14_ClN_3_O_3_S [M + H]^+^ 364.0517, found 364.0530.

*1-(Benzo[d]thiazol-2-yl)-3-(3-chloro-4-hydroxyphenyl)-1,3-dimethylurea* (**49**) Yield 65%; mp: 227–228.5 °C; ^1^H-NMR (500 MHz, DMSO-*d*_6_): δ (ppm) 10.09 (s, 1H), 7.74 (d, *J* = 7.7 Hz, 1H), 7.47 – 7.37 (m, 2H), 7.36 (d, *J* = 2.4 Hz, 1H), 7.23 (t, *J* = 7.9 Hz, 1H), 7.13 (dd, *J* = 8.6, 2.2 Hz, 1H), 6.95 (d, *J* = 8.7 Hz, 1H); ^13^C-NMR (126 MHz, DMSO-*d*_6_): δ (ppm) 165.20, 160.92, 150.53, 137.50, 136.58, 127.60, 126.56, 125.93, 125.38, 123.01, 122.40, 118.67, 115.86, 111.35, 37.16, 31.51; HRMS (ESI) calcd for C_16_H_14_ClN_3_O_2_S [M + H]^+^ 348.05680, found 348.05661.

*1-(3-Chloro-4-hydroxyphenyl)-3-(6-isopropylbenzo[d]thiazol-2-yl)urea* (**61**) Yield 43%; mp: 246–248 °C; ^1^H-NMR (500 MHz, DMSO-*d*_6_): δ (ppm) 10.83 (br s, 1H), 9.91 (br s, 1H), 9.02 (s, 1H), 7.74 (s, 1H), 7.61 (d, *J* = 2.4 Hz, 1H), 7.54 (d, *J* = 8.2 Hz, 1H), 7.26 (dd, *J* = 8.3, 1.5 Hz, 1H), 7.19 (dd, *J* = 8.7, 2.4 Hz, 1H), 6.93 (d, *J* = 8.7 Hz, 1H), 6.73 (d, *J* = 8.7 Hz, 1H), 2.97 (sept, *J* = 6.9 Hz, 1H), 1.23 (d, *J* = 6.9 Hz, 6H); ^13^C-NMR (126 MHz, DMSO-*d*_6_): δ (ppm) 159.33, 152.47, 148.89, 146.08, 143.46, 131.02, 130.84, 124.67, 120.74, 119.41, 119.33, 118.68, 116.64, 33.41, 24.17; HRMS (ESI) calcd for C_17_H_16_ClN_3_O_2_S [M + H]^+^ 362.0725, found 362.0721.

*1-(6-(Tert-butyl)benzo[d]thiazol-2-yl)-3-(3-chloro-4-hydroxyphenyl)urea* (**62**) Yield 66%; mp: 238–240 °C; ^1^H-NMR (500 MHz, DMSO-*d*_6_): δ (ppm) 9.75 (s, 1H), 7.90 (d, *J* = 1.9 Hz, 1H), 7.60 (d, *J* = 2.6 Hz, 1H), 7.56 (d, *J* = 8.5 Hz, 1H), 7.43 (dd, *J* = 8.5, 2.0 Hz, 1H), 7.18 (dd, *J* = 8.8, 2.6 Hz, 1H), 6.95 (d, *J* = 8.7 Hz, 1H), 1.32 (s, 9H); ^13^C-NMR (126 MHz, DMSO-*d*_6_): δ (ppm) 159.43, 152.17, 148.89, 145.88, 145.02, 130.87, 130.77, 123.74, 120.40, 119.36, 119.08, 118.41, 117.78, 116.74, 34.62, 31.45; HRMS (ESI) calcd for C_18_H_18_ClN_3_O_2_S [M + H]^+^ 376.0881, found 376.0882.

*1-(3-Chloro-4-hydroxyphenyl)-3-(6-ethoxybenzo[d]thiazol-2-yl)urea* (**63**) Yield 90%; mp: 255–257 °C; ^1^H-NMR (300 MHz, DMSO-*d*_6_): δ (ppm) 10.72 (br s, 1H), 9.92 (s, 1H), 9.00 (s, 1H), 7.60 (s, 1H), 7.57–7.36 (m, 2H), 7.18 (d, *J* = 7.9 Hz, 1H), 7.04 – 6.84 (m, 2H), 4.04 (q, *J* = 6.9 Hz, 2H), 1.33 (t, *J* = 7.0 Hz, 3H); ^13^C-NMR (75 MHz, DMSO-*d*_6_): δ (ppm) 157.76, 154.90, 152.19, 148.90, 141.98, 132.33, 130.82, 120.74, 119.89, 119.41, 119.34, 116.67, 114.76, 105.59, 63.59, 14.75; HRMS (ESI) calcd for C_16_H_14_ClN_3_O_3_S [M + H]^+^ 364.0517, found 364.0521.

*1-(3-Chloro-4-hydroxyphenyl)-3-(6-((trifluoromethyl)thio)benzo[d]thiazol-2-yl)urea* (**64**) Yield 24%; mp: 256–258 °C; ^1^H-NMR (500 MHz, DMSO-*d*_6_): δ (ppm) 11.04 (br s, 1H), 9.95 (br s, 1H), 9.03 (s, 1H), 8.37 (d, *J* = 1.8 Hz, 1H), 7.75 (d, *J* = 8.4 Hz, 1H), 7.67 (dd, *J* = 8.4, 1.9 Hz, 1H), 7.59 (d, *J* = 2.6 Hz, 1H), 7.19 (dd, *J* = 8.8, 2.6 Hz, 1H), 6.95 (d, *J* = 8.7 Hz, 1H); ^13^C-NMR (126 MHz, DMSO-*d*_6_): δ (ppm) 162.27, 152.10, 150.64, 149.08, 134.08, 132.73, 130.61, 130.28, 129.69 (q, *J* = 308.1 Hz), 120.65, 120.34, 119.38, 119.33, 116.74, 115.69 (q, *J* = 2.1 Hz); HRMS (ESI) calcd for C_15_H_9_ClF_3_N_3_O_2_S_2_ [M + H]^+^ 419.9850, found 419.9873.

*1-(3-Chloro-4-hydroxyphenyl)-3-(6-thiocyanatobenzo[d]thiazol-2-yl)urea* (**65**) Yield 80%; mp: 251–253 °C; ^1^H-NMR (500 MHz, DMSO-*d*_6_): δ (ppm) 11.04 (br s, 1H), 9.97 (br s, 1H), 9.13 (br s, 1H), 8.32 (d, *J* = 2.0 Hz, 1H), 7.74 (d, *J* = 8.5 Hz, 1H), 7.64 (dd, *J* = 8.5, 2.0 Hz, 1H), 7.60 (d, *J* = 2.6 Hz, 1H), 7.19 (dd, *J* = 8.8, 2.6 Hz, 1H), 6.94 (d, *J* = 8.7 Hz, 1H); ^13^C-NMR (126 MHz, DMSO-*d*_6_): δ (ppm) 161.91, 152.15, 149.47, 149.14, 133.11, 130.46, 129.54, 125.55, 121.00, 120.73, 119.66, 119.35, 116.75, 116.66, 112.20; HRMS (ESI) calcd for C_15_H_9_ClN_4_O_2_S_2_ [M + H]^+^ 376.9928, found 376.9939.

*1-(3-Chloro-4-hydroxyphenyl)-3-(6-(methylsulfonyl)benzo[d]thiazol-2-yl)urea* (**66**) Yield 78%; mp: 293–295 °C; ^1^H-NMR (500 MHz, DMSO-*d*_6_): δ (ppm) 11.16 (br s, 1H), 9.94 (br s, 1H), 9.25 (s, 1H), 8.55 (d, *J* = 1.7 Hz, 1H), 7.89 (dd, *J* = 8.5, 1.9 Hz, 1H), 7.82 (d, *J* = 8.5 Hz, 1H), 7.61 (d, *J* = 2.6 Hz, 1H), 7.20 (dd, *J* = 8.8, 2.6 Hz, 1H), 6.95 (d, *J* = 8.7 Hz, 1H), 3.23 (s, 3H); ^13^C-NMR (126 MHz, DMSO-*d*_6_): δ (ppm) 163.39, 152.08, 149.15, 134.69, 131.79, 130.45, 124.81, 121.71, 120.89, 119.56, 119.36, 116.68, 44.09; HRMS (ESI) calcd for C_15_H_12_ClN_3_O_4_S_2_ [M + H]^+^ 398.0031, found 398.0048.

*1-(3-Chloro-4-hydroxyphenyl)-3-(6-((trifluoromethyl)sulfonyl)benzo[d]thiazol-2-yl)urea* (**67**) Yield 94%; mp: 267–269 °C; ^1^H-NMR (300 MHz, DMSO-*d*_6_): δ (ppm) 11.40 (br s, 1H), 10.01 (s, 1H), 9.10 (br s, 1H), 8.89 (s, 1H), 8.02 (dd, *J* = 8.6, 1.9 Hz, 1H), 7.95 (d, *J* = 8.4 Hz, 1H), 7.60 (d, *J* = 2.5 Hz, 1H), 7.21 (dd, *J* = 8.7, 2.3 Hz, 1H), 6.95 (d, *J* = 8.8 Hz, 1H); ^13^C-NMR (75 MHz, DMSO-*d*_6_): δ (ppm) 166.05, 155.92, 149.35, 133.47, 130.23, 128.07, 126.29, 121.75, 121.17, 119.85, 119.36, 117.44, 116.67, 40.35, 40.08, 39.80, 39.52, 39.24, 38.96, 38.69; HRMS (ESI) calcd for C_15_H_9_ClF_3_N_3_O_4_S_2_ [M + H]^+^ 451.9748, found 451.9749.

*1-(3-Chloro-4-hydroxyphenyl)-3-(4,5,6,7-tetrahydrobenzo[d]thiazol-2-yl)urea* (**71**) Yield 58%; mp: 258–260 °C; ^1^H-NMR (500 MHz, DMSO-*d*_6_): δ (ppm) 9.91 (br s, 1H), 7.55 (d, *J* = 2.4 Hz, 1H), 7.14 (dd, *J* = 8.7, 2.4 Hz, 1H), 6.94 (d, *J* = 8.7 Hz, 1H), 2.59 (s, 2H), 2.54 (s, 2H), 1.75 (s, 4H); ^13^C-NMR (126 MHz, DMSO-*d*_6_): δ (ppm) 158.38, 151.01, 149.02, 138.46, 130.54, 120.47, 120.31, 119.38, 119.00, 116.77, 24.13, 22.46, 22.07, 21.82; HRMS (ESI) calcd for C_14_H_14_ClN_3_O_2_S [M + H]^+^ 324.05680, found 324.05634.

*1-(3-Chloro-4-hydroxyphenyl)-3-(2,3-dihydro-1H-inden-2-yl)urea* (**72**) Yield 21%; mp: 203–205 °C; ^1^H-NMR (500 MHz, DMSO-*d*_6_): δ (ppm) 9.59 (br s, 1H), 8.14 (br s, 1H), 7.52 (d, *J* = 2.6 Hz, 1H), 7.29 – 7.19 (m, 2H), 7.19 – 7.10 (m, 2H), 6.97 (dd, *J* = 8.7, 2.6 Hz, 1H), 6.83 (d, *J* = 8.7 Hz, 1H), 6.36 (d, *J* = 7.3 Hz, 1H), 4.48 – 4.31 (m, 1H), 3.25 – 3.09 (m, 2H), 2.84 – 2.67 (m, 2H); ^13^C-NMR (126 MHz, DMSO-*d*_6_): δ (ppm) 155.00, 147.38, 141.23, 132.93, 126.40, 124.56, 119.34, 119.16, 117.90, 116.55, 50.77, 39.70; HRMS (ESI) calcd for C_16_H_15_ClN_2_O_2_ [M + H]^+^ 303.08948, found 303.08908.

*1-(3-Chloro-4-hydroxyphenyl)-3-(4-(4-chlorophenyl)thiazol-2-yl)urea* (**73**) Yield 71%; mp: 206–208 °C; ^1^H-NMR (500 MHz, DMSO-*d*_6_): δ (ppm) 10.69 (s, 1H), 9.91 (s, 1H), 8.76 (s, 1H), 7.92 – 7.87 (m, 2H), 7.60 – 7.57 (m, 2H), 7.50 – 7.45 (m, 2H), 7.14 (dd, *J* = 8.7, 2.6 Hz, 1H), 6.93 (d, *J* = 8.7 Hz, 1H); ^13^C-NMR (126 MHz, DMSO-*d*_6_): δ (ppm) 159.31, 151.59, 148.87, 147.41, 133.16, 132.07, 130.69, 128.67, 127.26, 120.67, 119.34, 116.66, 107.89; HRMS (ESI) calcd for C_16_H_11_Cl_2_N_3_O_2_S [M + H]^+^ 380.00218, found 380.00168.

*1-(3-Chloro-4-hydroxyphenyl)-3-(thiazol-2-yl)urea* (**74**) Yield 74%; mp: 220–222 °C; ^1^H-NMR (500 MHz, DMSO-*d*_6_): δ (ppm) 9.83 (s, 1H), 9.81 (s, 1H), 7.56 (d, *J* = 2.6 Hz, 1H), 7.47 (dd, *J* = 3.9, 1.6 Hz, 1H), 7.20 (dd, *J* = 3.8, 1.5 Hz, 1H), 7.15 (dd, *J* = 8.8, 2.6 Hz, 1H), 6.95 (d, *J* = 8.7 Hz, 1H); ^13^C-NMR (126 MHz, DMSO-*d*_6_): δ (ppm) 160.49, 151.25, 149.00, 133.59, 130.64, 120.41, 119.39, 119.08, 116.78, 113.06; HRMS (ESI) calcd for C_10_H_8_ClN_3_O_2_S [M + H]^+^ 270.0099, found 270.0099.

*1-(3-Chloro-4-hydroxyphenyl)-3-(4-methoxyphenethyl)urea* (**75**) Yield 21%; mp: 161.5–163.5 °C; ^1^H-NMR (500 MHz, DMSO-*d*_6_): δ (ppm) 9.58 (br s, 1H), 8.29 (br s, 1H), 7.52 (d, *J* = 2.5 Hz, 1H), 7.14 (d, *J* = 8.5 Hz, 2H), 6.97 (dd, *J* = 8.7, 2.5 Hz, 1H), 6.89–6.84 (m, 2H), 6.82 (d, *J* = 8.7 Hz, 1H), 5.98 (t, *J* = 5.5 Hz, 1H), 3.72 (s, 3H), 3.33 – 3.19 (m, 2H), 2.66 (t, *J* = 7.2 Hz, 2H); ^13^C-NMR (126 MHz, DMSO-*d*_6_): δ (ppm) 157.65, 155.20, 147.32, 133.06, 131.35, 129.58, 119.37, 119.14, 117.90, 116.54, 113.78, 54.97, 40.87, 34.96; HRMS (ESI) calcd for C_16_H_17_ClN_2_O_3_ [M + H]^+^ 321.10005, found 321.09967.

*1-(Benzo[d]thiazol-6-yl)-3-(3-chloro-4-hydroxyphenyl)urea* (**76**) Yield 24%; mp: 224–226 °C; ^1^H-NMR (500 MHz, DMSO-*d*_6_): δ (ppm) 9.77 (s, 1H), 9.20 (s, 1H), 8.89 (s, 1H), 8.60 (s, 1H), 8.35 (d, *J* = 2.1 Hz, 1H), 7.97 (d, *J* = 8.8 Hz, 1H), 7.60 (d, *J* = 2.6 Hz, 1H), 7.47 (dd, *J* = 8.8, 2.2 Hz, 1H), 7.11 (dd, *J* = 8.7, 2.6 Hz, 1H), 6.90 (d, *J* = 8.7 Hz, 1H); ^13^C-NMR (126 MHz, DMSO-*d*_6_): δ (ppm) 153.69, 152.69, 148.26, 148.17, 137.64, 134.46, 131.92, 122.92, 120.22, 119.26, 118.82, 118.11, 116.63, 110.26; HRMS (ESI) calcd for C_14_H_10_ClN_3_O_2_S [M + H]^+^ 320.0255, found 320.0264.

*1,3-Bis(3-chloro-4-hydroxyphenyl)urea* (**78**) Yield 94%; mp: 254.5–256.5 °C; ^1^H-NMR (500 MHz, DMSO-*d*_6_): δ (ppm) 9.71 (br s, 2H), 8.43 (br s, 2H), 7.54 (d, *J* = 2.6 Hz, 2H), 7.06 (dd, *J* = 8.8, 2.6 Hz, 2H), 6.87 (d, *J* = 8.7 Hz, 2H); ^13^C-NMR (126 MHz, DMSO-*d*_6_): δ (ppm) 152.77, 147.99, 132.14, 120.11, 119.22, 118.71, 116.59; HRMS (ESI) calcd for C_13_H_10_Cl_2_N_2_O_3_ [M + H]^+^ 313.01412, found 313.01361.

### 3.3. β-HSD10 Enzymatic Activity Assay

Purification of 17β-HSD10 protein was performed as described in our previous work [22].

Compound screening, dose response and mechanism of inhibition experiments were performed as described in Hroch et al. 2016, with the exceptions being an alteration in assay buffer (10 mM Tris HCl (pH 7.4), 150 mM NaCl, 1 mM DTT, 0.005% Tween20, 0.01% BSA) and a change in temperature to 25 °C, to improve enzyme stability and assay reliability as compound solubility improved, as outlined in our previous work [19]. 

Orthogonal screening (small molecule aggregation and redox cycling experiments) was carried out as described in our previous work [19].

### 3.4. Lactate Dehydrogenase (LDH) Cytotoxicity Assay

Cell cytotoxicity was assessed via the measurement of lactate dehydrogenase leakage into the culture medium using a commercially available kit from Pierce (Thermo Scientific, UK, cat no. 88953). This was carried out in accordance with the kit guidelines, with the activity of LDH being calculated from the change in absorbance at 340 nm as NADH is reduced. HEK293 cells overexpressing mts17β-HSD10 were cultured in phenol-red free media (10% FBS, 1 mM Sodium Pyruvate, 100 units Penicillin, 0.1 mg/mL Streptomycin and 2 mM L-Glutamine) and seeded at a density of 10,000 cells per well (100 µL, 96-well plates). Cells were then treated with compound of interest at 2 concentrations (25 µM and 100 µM in DMSO) in triplicate. Treated cells were then incubated at 37 °C and CO_2_ (5%) for 24 hours before the LDH assay was performed as per the manufacturer’s instructions. Spontaneous control (water) and maximum control (lysis buffer) used in accordance with the kit guide. Absorbance was measured at 490 nm and 680 nm using the SpectraMaxM2e spectrophotometer (Molecular Devices, San Jose, CA, USA). The measured LDH activity was used to calculate % cytotoxicity using the following equation: (1)$$\%\;\mathrm{cytotoxicity}=\frac{\left(\mathrm{compound}\;\mathrm{treated}\;\mathrm{LDH}\;\mathrm{Activity}-\mathrm{Spontaneous}\;\mathrm{LDH}\;\mathrm{Activity}\right)}{\left(\mathrm{Maximum}\;\mathrm{LDH}\;\mathrm{Activity}-\mathrm{Spontaneous}\;\mathrm{LDH}\;\mathrm{Activity}\right)}\times 100$$

### 3.5. (−)-CHANA Assay – In Vitro Dose Response and EC_50_ Determination

HEK293 mts17β-HSD10 cells were seeded at a density of 10,000 cells per well (100 µL, 96-well black plates, Greiner Cat no. 655090) in phenol-red free media (10% FBS, 1 mM Sodium Pyruvate, 100 units Penicillin, 0.1 mg/mL Streptomycin and 2 mM L-Glutamine). The media was removed from the cells and replaced with fresh media containing varying concentrations of compound (100 µM–0.098 µM). The fluorogenic probe (−)-CHANA was then added to each well to give a final assay concentration of 20 µM. Fluorescence was immediately measured using the FLUOstar Optima microplate reader (excitation = 380 nm, emission = 520 nm, orbital averaging = 3 mm) and the initial reaction monitored for 3–4 hours. EC_50_ was calculated from the control - subtracted triplicates using non-linear regression (four parameters) of GraphPad Prism 5 software. Final EC_50_ and SEM value was obtained as a mean of at least 3 independent measurements.

## 4. Conclusions

In summary, four novel series of benzothiazolylureas were designed and synthesised. All compounds were evaluated for 17β-HSD10 inhibitory ability in vitro, where compounds **5**, **6**, and **63** showed the most promising 17β-HSD10 inhibitory activity in our enzymatic assays, although the orthogonal screens appear to indicate that **63** could be inhibiting 17β-HSD10 in an unfavourable manner. Key structure–activity relationships have been established and further validated with a urea linker and a 4-phenolic moiety with a 3-halogen substitution confirmed to be essential for compound 17β-HSD10 inhibitory ability. Furthermore, a bulky substitution (e.g., *t*-butyl) in position 6 of the benzothiazole moiety appears to be the most promising, potentially occupying the chemical space more effectively within the binding site. 

Positively the most promising compounds were also shown to have an inhibitory effect at a cellular level with limited cytotoxicity and all hit compounds display a more favourable kinetic mechanism of action (reversible mixed inhibition) to other previously published work. 

These findings provide significant structural activity insight into our 17β-HSD10 inhibitor compound design and are our most promising observations to date. With further hit optimisation and neuronal cellular evaluation to determine if these compounds are protective against Aβ-mediated cytotoxicity, this could potentially lead to novel class of therapeutics for AD.

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
