# Peer review of "Novel Benzothiazole-Based Ureas as 17β-HSD10 Inhibitors, A Potential Alzheimer’s Disease Treatment"

_molecules, 2019, doi:10.3390/molecules24152757_

Round 1
Reviewer 1 Report
The article is of considerable interest as the development of a fairly promising direction in the search for drugs to treat Alzheimer's disease. Evaluation of compounds as inhibitors was carried out both in the in vitro system and on cells. I would like a more detailed discussion of the grounding and importance of Orthogonal Counter Screens, and at least for leading compounds assessing their cytoprotective activity under amyloid toxicity conditions.
Author Response
Dear Reviewer 1,
Firstly we would like to thank you for your feedback and very positive review. To address the two points that you have made:
1) I would like a more detailed discussion of the grounding and importance of Orthogonal Counter Screens.
The grounding and importance of orthogonal counter screens has been heavily covered in our previous referenced assay development paper and as such we have not gone into further discussion on this. We feel that the level of detail and explanation we have given is acceptable in the scope of this paper.
2) At least for leading compounds assessing their cytoprotective activity under amyloid toxicity conditions
Assessing their cytoprotective nature under amyloid toxicity conditions will be carried out at a later date and is out with the scope of this study. However, we have included a sentence in our closing statement to highlight that this has not been overlooked and will be assessed at a later date.
Reviewer 2 Report
Aitken et al. report on new 17beta-HSD10 inhibitors, which might be useful as a therapeutic approach targeting disease-relevant mechanisms. The paper is interesting and well written. The methods are appropriate and all relevant information is given. However there are some points, which need attention:
1. It is claimed in line 45/46 that the authors (and others) “have shown that inhibition of this enzyme is beneficial in AD”. I do not see that the references provide data from clinical studies with AD patients, which would justify such a strong statement. It is also mentioned that it “protects against Abeta toxicity” – are these animal studies or just experiments using cultured cells and in vitro experiments; this needs to be clarified.
2. There are other statements where it is claimed that these are approaches “in treating AD” (e.g., line 67 and 72). It would be more appropriate to phrase it in a way that these approaches may target potentially disease-relevant mechanisms.
3. Line 86: “to also generate compounds with improved tolerance” – this statement needs to be explained.
4. Line 257 and 259: reference to Table 7 and Table 8 – I do not see the respective tables in the manuscript.
Author Response
Dear Reviewer 2,
Firstly we would like to thank you for your feedback and very positive review. To address the four points that you have made:
1. It is claimed in line 45/46 that the authors (and others) “have shown that inhibition of this enzyme is beneficial in AD”. I do not see that the references provide data from clinical studies with AD patients, which would justify such a strong statement. It is also mentioned that it “protects against Abeta toxicity” – are these animal studies or just experiments using cultured cells and in vitro experiments; this needs to be clarified.
We agree with the reviewer that this sentence was a little clumsy and did need clarification and we have changed the sentence to "Importantly, we and others have shown that inhibition of this enzyme is beneficial in both in vitro and in vivo AD models in its own right and also protects against Aβ toxicity in both cellular and transgenic mouse models of AD (Lim et al. 2011; Valasani et al. 2014; Hroch et al. 2016; Hroch et al. 2017; Benek et al. 2017; Xiao et al. 2019).
We have also included and referenced Xiao et al 2019, the most recent paper recognising our protein as a potential drug target in AD, in support of this statement.
2. There are other statements where it is claimed that these are approaches “in treating AD” (e.g., line 67 and 72). It would be more appropriate to phrase it in a way that these approaches may target potentially disease-relevant mechanisms.
We have removed "in treating AD" and changed this to "may target potential disease- relevant mechanisms" as the reviewer has suggested.
3. Line 86: “to also generate compounds with improved tolerance” – this statement needs to be explained.
We have altered this statement to; "to also generate compounds with improved tolerance and less cytotoxicity within our cellular assays" to explain this further.
4. Line 257 and 259: reference to Table 7 and Table 8 – I do not see the respective tables in the manuscript.
This is a typographical error which we have corrected within the manuscript.
Reviewer 3 Report
In the current manuscript the authors present the synthesis and evaluation of novel bezothiazole-based molecules with inhibitory activity over 17B-HSD10, a mitochondrial protein which interacts with AB and is associated witht he pathophysiology of AD.
the manuscript is well written and sound. I have only minor comments:
Ln 35-36: the word "cycle" is repeated
ln 45-46: Auhtors should better reconcile the contradictory evidence that 17B-HDS10 activity is important in AD due to decay in glucose metabolism and the protective effect afforded by inhibition
Ln 322 - Don't the authors mean 61% increase (and not decrease as is written) in 17B-HDS10 activity in the presence of Triton x-100?
Author Response
Dear Reviewer 3,
Firstly we would like to thank you for your feedback and very positive review. To address the three points that you have made:
1) Ln 35-36: the word "cycle" is repeated
This typographical error has been removed from the manuscript.
2) ln 45-46: Auhtors should better reconcile the contradictory evidence that 17B-HDS10 activity is important in AD due to decay in glucose metabolism and the protective effect afforded by inhibition
We have added an additional sentence to further clarify the importance of 17β-HSD10 inhibition.
"A current working hypothesis is that by inhibiting the enzyme activity of 17β-HSD10 (a contributor to the β-fatty acid oxidation pathway) this can re-balance alterations in glucose metabolism observed in AD (Aitken unpublished data)".
3) Ln 322 - Don't the authors mean 61% increase (and not decrease as is written) in 17B-HDS10 activity in the presence of Triton x-100?
Yes increase not decrease, this typographical error has been removed in the manuscript.